# Internet-Based Psychological Interventions during SARS-CoV-2 Pandemic: An Experience in South of Italy

**DOI:** 10.3390/ijerph19095425

**Published:** 2022-04-29

**Authors:** Grazia D’Onofrio, Filomena Ciccone, Giuliana Placentino, Maria Placentino, Cinzia Tulipani, Annamaria Prencipe, Gabriella De Vincentis

**Affiliations:** 1Clinical Psychology Service, Health Department, Fondazione IRCCS Casa Sollievo della Sofferenza, San Giovanni Rotondo, 71013 Foggia, Italy; f.ciccone@operapadrepio.it (F.C.); giuliana.placentino@operapadrepio.it (G.P.); maria.placentino78@gmail.com (M.P.); cinziatulipani09@gmail.com (C.T.); am.prencipe@operapadrepio.it (A.P.); 2Health Department, Fondazione IRCCS Casa Sollievo della Sofferenza, San Giovanni Rotondo, 71013 Foggia, Italy; g.devincentis@operapadrepio.it

**Keywords:** SARS-CoV-2, internet-based support, post-traumatic stress disorder, mental health

## Abstract

Severe Acute Respiratory Syndrome Coronavirus 2 (SARS-CoV-2) has led to an increasing demand for online psychological intervention. The aim of this study is to evaluate the efficacy of received support in internet-based psychological intervention group (I-IG) patients, compared with a wait-list control group (CG). The Impact of Event Scale—Revised, Patient Health Questionnaire 9-item and Generalized Anxiety Disorder scale 7-item were administered. After participants had used the internet-based solution, the System Usability Scale was administered. In total, 221 patients (194 patients supported by internet-based interventions and 27 patients supported onsite) were included in intervention group, and 194 patients were included in CG. In a 6-month follow-up, participants in the I-IG demonstrated significant improvements in terms of PTSD risk (*p* < 0.0001, *d* = 0.64), depression (*p* < 0.0001, *d* = 0.68), and anxiety (*p* < 0.0001, *d* = 1.33), compared to the CG. Significant improvements in onsite intervention group patients with a large to very large effect size of PTSD risk (*p* < 0.0001, *d* = 0.91), depression (*p* < 0.0001, *d* = 0.81), and anxiety (*p* < 0.0001, *d* = 1.62) were found. After internet-based solution use, I-IG patients reported a very high usability and functionality (72.87 ± 13.11) of online intervention. In conclusion, SARS-CoV-2-related mental health problems can be improved by internet-based psychological intervention. The usability and functionality evaluation of online solutions by technological tools showed very positive results for the I-IG patients.

## 1. Introduction

At the end of 2019, a novel Severe Acute Respiratory Syndrome Coronavirus 2 (SARS-CoV-2) emerged worldwide. In 2020, in Italy, the first outbreak of the SARS-CoV-2 pandemic was acknowledged. As a consequence of the first deaths, quarantines and restrictive measures, until the lockdown, were put into place nationally.

With the increase in hospitalized SARS-CoV-2 cases, there was excessive use of hospital beds, particularly in Italy, underlining that past reductions in the number of beds in many hospitals in the country [1,2,3].

This reduction, not only limited to Italy, is mainly related to the progressive cuts in healthcare spending at the national and local levels, which led to an unbalanced number of beds among the population [4].

In this scenario, self-isolation, social distancing, and quarantines further augmented the distress, with a following increasing demand for psychological intervention.

In current studies, it has been shown that psychological interventions have a benefit on mental health outcomes, even if conducted online, in particular with a larger effect size on the mitigation of mental health symptoms and increasing resilience [5,6,7].

Psychological interventions have a positive impact on symptoms such as depression, anxiety and perceived stress in healthcare workers and patients. American Psychological Association and International Society of Traumatic Stress Studies specified to constantly monitor the mental health status of pandemic-affected populations and provide timely evidence-based trauma-centered psychotherapies [8,9,10]. 

However, beyond the above, the importance of enhancing internet-based psychological support emerges, because of traditional face-to-face psychological facilities unavailable during the outbreak [10]. Tele-medical approaches are needed to be implemented in public health strategies in order to support many people simultaneously and effectively [11]. The benefits of e-mental health approaches are great and innovative resources and currently more necessary than ever before [12]. Bäuerle et al. reported that a developed e-mental health intervention, “CoPE It”, offered an evidence-based psychotherapeutic/psychological support to overcome psychological distress in times of SARS-CoV-2 [12]. 

Given the evidence for the risk of short- and long-term psychological consequences and their impact on job performance and quality of care [13], psychological packages should be urgently implemented to manage mental healthcare for medical staff working in the frontline against pandemic [14] and for the general population. 

In light of the SARS-CoV-2 emergency, we developed an online help desk, “In Ascolto” (“Tune in”), in the Clinical Psychological Service of Casa Sollievo della Sofferenza hospital.

In order to evaluate the efficacy of received support, a case–control study has been designed with a randomized waiting list as the control group. It has been hypothesized that there would be a major containment of post-traumatic stress, and a decrease in depression and anxiety, compared with the control group.

## 2. Materials and Methods

The present study was conducted according to the Declaration of Helsinki, the Guidelines for Good Clinical Practice and the Strengthening the Reporting of Observational Studies in Epidemiology (STROBE) guidelines [15]. This is a case–control investigation, in which the assignment of an intervention to the participants, its effect assessment and health-related behavioral outcomes are considered. In the present study, randomized waiting list patients were recruited as control subjects.

### 2.1. Study Sample

From February 2020 to January 2021, we enrolled people who contacted the Clinical Psychology Service of Istituto di Ricovero e Cura a Carattere Scientifico (IRCCS) “Casa Sollievo della Sofferenza” and received a psychological intervention. The control group patients were put on a waiting list for psychological support. We obtained written informed consent for research from each patient (parents signed for minors). All subjects were Caucasian, not including people of Jewish, Eastern European, nor North African descent, with most individuals having southern Italian ancestry, living in southern Italy for at least three generations.

The inclusion criteria were: (1) laboratory-confirmed SARS-CoV-2 infection, diagnosed with mild-to-moderate [non-pneumonia/mild pneumonia, blood oxygen saturation >93% without oxygen support] [16,17,18] and/or (2) daily worry about SARS-CoV-2 and (3) at least one of the following criteria: impaired concentration and/or sleep problems and/or excessive news and social media usage and/or impaired work capacity and/or difficulties enjoying everyday activities. The exclusion criteria were: (1) diagnosed with critical (respiratory failure, septic shock, and/ormultiple organ dysfunction or failure) SARS-CoV-2 [17]; (2) having a history of psychotic disorders.

### 2.2. Outcomes

For all participants, a mental health assessment was performed at baseline and after six months of follow-up. 

The post-traumatic stress level was evaluated by Impact of Event Scale-Revised (IES-R) [19], a 22-item self-report measure that assesses subjective distress caused by traumatic events. Items were rated on a 5-point scale ranging from 0 (“not at all”) to 4 (“extremely”). The IES-R yields a total score (ranging from 0 to 88). A total IES-R score of 33 or over signifies the likely presence of post-traumatic stress disorder (PTSD).

Depression symptoms were assessed using the Patient Health Questionnaire 9-item (PHQ-9) [20], a depression module, which scores each of the nine items as “0” (not at all) to “3” (nearly every day); it is used to monitor the severity of depression and response to treatment. It can be used to make a tentative diagnosis of depression in at-risk populations. PHQ-9 score ≥10 had a sensitivity of 88% and a specificity of 88% for major depression.

Anxiety was evaluated by the Generalized Anxiety Disorder 7-item (GAD-7) [21], a self-administered 7-item instrument to identify probable cases of general anxiety disorder along with measuring anxiety symptom severity. It can also be used as a screening measure of panic, social anxiety, and PTSD. It was modeled after the PHQ-9 to be used quickly and effectively within a primary care setting. Scores of 5, 10, and 15 were taken as the cut-off points for mild, moderate and severe anxiety, respectively.

After the participants had used the internet-based solution, System Usability Scale (SUS) [22] was administered in order to assess the usability and functionality of the online solution and technological tools (smartphone and tablet for telephone and/or video-call) by participants.

### 2.3. Statistical Analyses

For continuous variables, normal distribution was verified by the Shapiro–Wilk normality test and the one-sample Kolgomorov–Smirnov test. For normally distributed variables, hypotheses regarding differences among the groups were compared by means of the Welch two sample *t*-test. For non-normally distributed variables, hypotheses regarding differences among the groups were compared by means of the Wilcoxon rank sum test with continuity correction or by means of the Kruskal–Wallis rank sum test.

For dichotomous variables, hypotheses regarding differences between the groups were tested using a chi-square test. This analysis was conducted using the 2-Way Contingency Table Analysis. 

Risks (adjusted by age) are reported as odds ratios (OR) along with their 95% confidence intervals (CI). All the statistical analyses were conducted with the R Ver. 2.8.1 statistical software package (the R Project for Statistical Computing; available at URL http://www.r-project.org/ (accessed on 20 July 2021)). Tests in which the *p* value was smaller than the type I error rate α = 0.05 were declared significant.

## 3. Results

As shown in Figure 1, during the enrolment period, 422 people contacted the Clinical Psychology Service for support. Of these, seven patients were excluded because they were affected by critical SARS-CoV-2. Thus, the final population included 415 patients, 121 men (29.20%) and 294 women (70.80%), with a mean age of 46.31 ± 14.29 years and a range from 13 to 90 years. Therefore, 221 patients (M: 67; F: 154; mean age of 46.03 ± 14.78 years; range: 13–90 years) were included in the intervention group, and 194 patients (M: 54; F: 140; mean age of 46.42 ± 13.76 years; range: 15–90 years) were included in the waitlist control group (CG).

As explained in Table 1, the intervention group patients were split into two groups: (1) 194 patients supported by an internet-based psychological intervention (I-IG), and (2) 27 patients supported by onsite psychological intervention (O-IG), i.e., face to face in hospital wards dedicated to SARS-CoV-2. 

The intervention group and CG did not differ in sex distribution (*p* = 0.579), age (*p* = 0.779), educational level (*p* = 0.682), role (*p* = 0.728), provenance (*p* = 0.115), previous psychological disorders (*p* = 0.982), and previous psychological consultation (*p* = 0.887).

As shown in Table 2, after 6 months of follow-up, there were significant improvements in I-IG patients with a medium effect size for PTSD risk, depression, and anxiety severity: (1) IES-R: improvement of 13.46%, *p* < 0.0001, and *d* = 0.64; (2) PHQ-9: improvement of 4.71%, *p* < 0.0001, and *d* = 0.68; and (3) GAD-7: reduction of 6.69%, *p* < 0.0001, and *d* = 1.33. 

After 6 months of follow-up, there were significant improvements in O-IG patients with a large to very large effect size for PTSD risk, depression, and anxiety severity: (1) IES-R: improvement of 17.33%, *p* < 0.0001, and *d* = 0.91; (2) PHQ-9: improvement of 5.45%, *p* < 0.0001, and *d* = 0.81; and (3) GAD-7: reduction of 7.11%, *p* < 0.0001, and *d* = 1.62.

In CG patients, after 6 months of follow-up, no significant improvements were reported (IES-R: *p* = 0.564; PHQ-9: *p* = 0.180; GAD-7: *p* = 0.109).

As shown in Table 3, usability and functionality of I-IG evaluated by SUS was shown. The patients confirmed a high level of usability of the technological solution (SUS > 70 in mean).

## 4. Discussion

In the present study, using a relatively large sample of patients in the south of Italy, it was illustrated that some SARS-CoV-2-related mental health problems can even be improved by internet-based psychological interventions with a medium effect size. 

These outcomes are in line with current studies. Many studies have shown small to very large Cohen’s d effect sizes, reporting the decreasing of depression and anxiety [23,24,25,26]. In other three study, the partial eta squared (η^2^p) effect size was calculated, revealing a significant large effect (η^2^p = 0.20–0.89) of condition on mental health difficulties with a decreasing of depression and anxiety symptoms [27,28,29], and traumatic symptoms [28]. A study reported, using a post hoc analyses of the individual time points, showed that depression and anxiety were significantly decreased (depression: F = 37.35, *p* < 0.001; anxiety: F = 26.58, *p* < 0.001), as well as a main effect of group (depression: F = 4.384, *p* = 0.047; anxiety: F = 5.634, *p* = 0.026) and a group-by-time interaction (depression: F = 5.268, *p* = 0.009; anxiety: F = 3.743, *p* = 0.031) [30].

Moreover, our study reported that internet-based psychological interventions yield good results even if not at the level of onsite psychological interventions.

We demonstrated that online psychological interventions can be a feasible and useful way to provide support to individuals during the SARS-CoV-2 crisis, diminishing the infection risk in mental healthcare providers [30,31]. Another benefit is that they are more accessible, independent with regard to time and place, with a high level of autonomy and privacy, and lower costs [32].

An innovative approach used in this study is the usability and functionality evaluation of the online solution by using technological tools (smartphone and tablet for telephone and/or video-call), which reported very high values in the I-IG patients. These results could be compared to the results of other studies [33,34] that reported high user satisfaction scores for online health services.

In Italy, the crisis that followed the SARS-CoV-2 emergency had a profound impact on the economic and social system. It is a completely new situation that has increased the number of poor families and individuals in our country: the number of people in absolute poverty went from an estimated 4.6 million in 2019 to 5.6 million [35].

The first wave of the pandemic had northern Italy as its epicenter. However, the economic crisis soon spread to southern Italy, where it translated into a social emergency more dramatically, crossing a weaker productive fabric, a more fragmented world of work and a more fragile society.

Therefore, the pandemic has put a strain on the psychological health of the population, especially in southern Italy, and over time the consequences have only emerged with greater insistence. SARS-CoV-2 has amplified previous conditions of discomfort and the prolonged stress has caused new ones. In light of this information, psychological intervention can intercept the problems of individuals, families and communities before they turn into more serious disorders up to chronicity, and determine well-being and a better quality of life.

Our findings have shown that it is possible to psychologically support people even online during a pandemic. Clearly, further studies should be carried out to extend the results to other settings such as school and sport, and other Italian and/or European geographical areas.

Presumably, when the pandemic is over, the normality to which one will return will coincide with a new way of working. The new modes will not replace tout court the classic setting, but they will be integrated into it.

It will therefore be necessary to assess case by case if the request to remain online can be evolutionary or defensive. The greater integration of this instrument should be assessed on a case-by-case basis, considering who can benefit clinically and who cannot, also with regard to any management entirely at a distance. This is a subject of interest, which is worth continued research.

Some limitations of the present study must be acknowledged. The study population comprised only Caucasian patients recruited in a single center, so it could be possible that our results may not be applicable to other populations. Furthermore, due to the urgency brought on by the epidemic, we could not run follow-ups at 1 and 3 months.

In conclusion, to our knowledge, studies that have examined online psychological intervention effectiveness for improving PTSD, depression and anxiety symptoms related to SARS-CoV-2 did not fully consider evaluating the usability of this technology. Our internet-based psychological intervention showed comparably higher rates of engagement, compliance, and tolerance among users when compared with onsite psychological intervention.

Future improvements include enhanced scene variety and extended intervention follow-up for stronger long-term effects on mental health outcomes.

## Figures and Tables

**Figure 1 ijerph-19-05425-f001:**
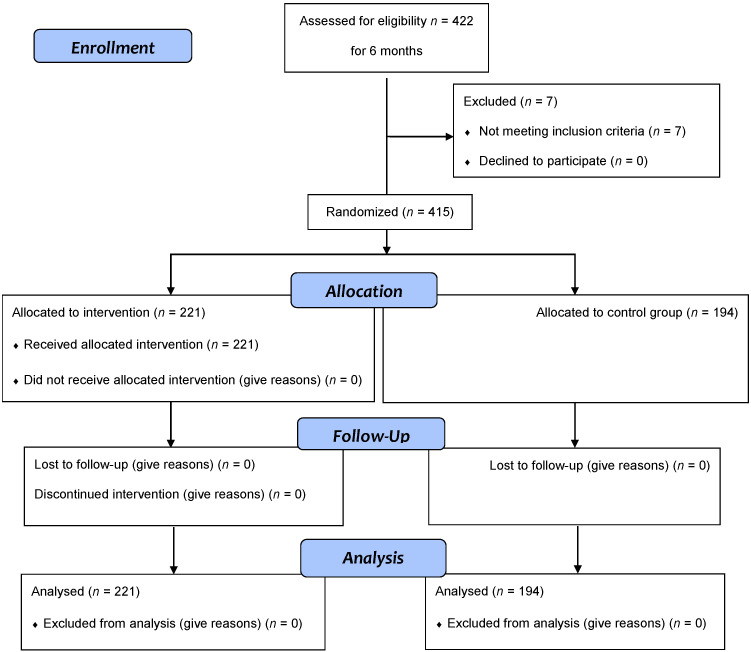
Flowchart of intervention.

**Table 1 ijerph-19-05425-t001:** Characteristics of people who contacted the help desk “In Ascolto” of the Clinical Psychological Service.

	All N = 415	Intervention Group *n* = 221	Control Group *n* = 194	*p*-Value
**Sex**				0.579
Male/Female	294/121	154/67	140/54	
Males (%)	29.20	30.3	27.8	
**Age (years)**				0.779
Mean ± SD	46.21 ± 14.29	46.03 ± 14.78	46.42 ± 13.76	
Range	13–90	13–90	15–90	
**Educational level (in years)**				0.682
Mean ± SD	13.73 ± 4.77	13.64 ± 4.80	13.84 ± 4.75	
Range	5–18	5–18	5–18	
**Role**				0.728
Patient—*n* (%)	265 (63.9)	145 (65.6)	120 (61.9)	
Health worker—*n* (%)	132 (31.8)	67 (30.3)	65 (33.5)	
Caregiver—*n* (%)	18 (4.3)	9 (4.1)	9 (4.6)	
**Provenance**				0.115
COVID ward—*n* (%)	156 (37.6)	78 (35.3)	78 (40.2)	
NoCOVID ward—*n* (%)	11 (2.7)	9 (4.1)	2 (1.0)	
Outside—*n* (%)	248 (59.8)	134 (60.6)	114 (58.8)	
**Previous psychological disorders**				0.982
Yes—*n* (%)	64 (15.4)	34 (15.4)	30 (15.5)	
No—*n* (%)	351 (84.6)	187 (84.6)	164 (84.5)	
**Previous psychological consultation**				0.887
Yes—*n* (%)	112 (27.0)	59 (26.7)	53 (27.3)	
No—*n* (%)	303 (73.0)	162 (73.3)	141 (72.7)	
**Setting**				-
Online (Phone)—*n* (%)	-	137 (62.0)	0	
Online (Videocall)—*n* (%)	-	57 (25.8)	0	
Onsite—*n* (%)	-	27 (12.2)	0	
**Number of sessions**				-
Mean ± SD	-	8.58 ± 7.23	0	
Range	-	1–35	0	

**Table 2 ijerph-19-05425-t002:** PTSD risk, depression and anxiety assessment of internet-based intervention group (I-IG), onsite intervention group (O-IG) and control group at baseline and after a 6-month follow-up.

	I-IG				O-IG				Control Group			
	Baseline	Follow-Up	*p*-Value	Cohen’s *d*	Baseline	Follow-Up	*p*-Value	Cohen’s *d*	Baseline	Follow-Up	*p*-Value	Cohen’s *d*
**IES-R**							**<0.0001**					
Mean ± SD	43.29 ± 23.33	29.83 ± 18.18	**<0.0001**	0.64	48.76 ± 21.57	31.43 ± 16.19		0.91	43.25 ± 24.46	43.26 ± 24.47	0.564	−0.0004
Range	5.50–74.00	3.60–68.00			7.12–68.00	4.80–60.00			5.50–74.00	5.50–75.00		
**PHQ-9**							**<0.0001**					
Mean ± SD	15.16 ± 6.93	10.45 ± 6.86	**<0.0001**	0.68	15.19 ± 6.74	9.74 ± 6.67		0.81	14.11 ± 8.41	14.13 ± 8.44	0.180	−0.002
Range	5–26	1–26			6–25	1–20			4–27	4–28		
**GAD-7**												
Mean ± SD	14.96 ± 4.98	8.27 ± 5.11	**<0.0001**	1.33	15.04 ± 4.55	7.93 ± 4.19	**<0.0001**	1.62	10.61 ± 6.29	10.65 ± 6.34	0.109	−0.006
Range	5–23	1–18			7–20	1–16			1–26	1–27		

*Legend*. **IES-R**, Impact of Event Scale-Revised; **PHQ-9**, Patient Health Questionnaire 9-item; **GAD-7**, Generalized Anxiety Disorder 7-item.

**Table 3 ijerph-19-05425-t003:** Usability and functionality of internet-based psychological intervention.

	I-IG N = 194
**SUS**	
Mean ± SD	72.87 ± 13.11
Range	45.00–100.00

*Legend.***SUS**, System Usability Scale.

## Data Availability

Not applicable.

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
