# Peer review of "Internet-Based Psychological Interventions during SARS-CoV-2 Pandemic: An Experience in South of Italy"

_ijerph, 2022, doi:10.3390/ijerph19095425_

Round 1

Reviewer 1 Report

Review of “Internet-based Psychological Interventions during SARS-CoV-2 pandemic: an Experience in South of Italy”

The paper explores the effectiveness of internet-based psychological interventions (i.e., by audio-call and by video-call) in a specific Italian geographical area and provide potential scientific support for this kind of treatment. I would encourage the authors to provide a rationale for the choice of this geographical area and expand the discussion including how these findings can be extended to other settings (e.g., educational, sporting, organizational), situations (e.g., other situations that can be compared to the COVID pandemic period), and contexts (are these findings transferable to other Italian or European geographical areas?).

Aside of these potential strengths of the Manuscript, I feel like major revisions and/or clarification are needed to make the paper suitable for publication on the “International Journal of Environmental Research and Public Health”. Starting from the Abstract: if the title is Internet-based psychological interventions, why 27 participants were supported in person? I think the authors should consider the sole sample of 194 patients supported by internet-based intervention and then run again the analyses, otherwise the rationale for the study is lost.

Personally, I would suggest relabelling the “IbIG” in “IG” and check for consistency throughout the Manuscript. The reader understands from the title that the intervention is internet-based and then the text becomes easier to read. Another aspect: If I rightly understand, the CG is a wait-list control group. This is very ethical and it is a strength of the paper. I would suggest stating this since the beginning in the abstract, i.e., “wait-list control group (CG)”, and being consistent throughout the Manuscript.

A main aspect is the following: It is unclear if a mental health assessment was performed at baseline. Is there a T0 in the study, or the authors just assessed mental health outcomes at the end of the six months? In the case there was an assessment at T0 and then at T1, the T-tests chosen for the analyses might not be the best option, and Repeated Measures ANOVAs/MANOVAs might be a better solution. Please, clarify.

Figures and Tables are well-formatted and presented. Despite the paper is well written, I suggest having it revised by a native English speaker, as they could help make the paper easier to read.

Major revisions:

Abstract

Lines 17-19: “In a 6-month follow-up, participants in the IG demonstrated significant improvements in terms of PTSD risk (…), depression (…), and anxiety (…) if compared to the CG.

Lines 19-21: I would remove this sentence and these analyses, due my concern about the fact that these participants are not supported via internet, but rather in person.

Keywords

The authors included some keywords that are already in the title, of course they are free to choose their keywords, but I would suggest excluding words already present in the title and include other concepts, e.g., stress, post traumatic stress disorder, ptsd, mental health.

Introduction

Line 54-55: Could the authors provide an English translation for “In Ascolto” in brackets?

Materials and methods

Study sample: The authors did not describe the sample in terms of age range, mean age and standard deviation, gender, … Please, provide further details.

Lines 72-75: What this background add to our scientific knowledge? Please, outline this point.

Statistical analyses

The description of statistical analyses is not in a logical order. I think the authors should start with:

  • Descriptive analyses of the study sample (e.g., age, gender frequency), then the authors do not need to report this step in “Statistical analyses”
  • Analysis of distribution of the outcome variables, and report this at the beginning of “Statistical analyses”, as normally distributed data allow for further parametric tests to be undertaken.
  • Then the authors choice of T-tests is well described and suitable for this kind of analysis.

Discussion

Lines 163-172: This paragraph should be included in the results section.

Generally, the discussion should be expanded.

Minor revisions:

Abstract

Lines 10-11: “an increasing demand”

Line 14: missing space “7-item were”

Line 15: remove been “was administered”

Line 23: please, remove “even”, “can be improved by internet-based psychological intervention”

Introduction

Line 29: please, remove “virus”, “a novel Severe Acute Respiratory Syndrome Coronavirus 2 (SARS-CoV-2) emerged worldwide.”

Materials and methods

Lines 98-99: please, add spaces “anxiety disorder along with…”

I hope this review will be hlepful for the authors to further improve their paper.

Best regards.

Author Response

  • The paper explores the effectiveness of internet-based psychological interventions (i.e., by audio-call and by video-call) in a specific Italian geographical area and provide potential scientific support for this kind of treatment. I would encourage the authors to provide a rationale for the choice of this geographical area and expand the discussion including how these findings can be extended to other settings (e.g., educational, sporting, organizational), situations (e.g., other situations that can be compared to the COVID pandemic period), and contexts (are these findings transferable to other Italian or European geographical areas?). Aside of these potential strengths of the Manuscript, I feel like major revisions and/or clarification are needed to make the paper suitable for publication on the “International Journal of Environmental Research and Public Health”.
  • We thank the reviewer for his/her positive and constructive observations. Below, please find item-by-item responses to your comments, which are included verbatim. In Discussion section we provided a rationale for the choice of this geographical area and expand the discussion including how these findings can be extended to other settings and contexts.

  • Starting from the Abstract: if the title is Internet-based psychological interventions, why 27 participants were supported in person? I think the authors should consider the sole sample of 194 patients supported by internet-based intervention and then run again the analyses, otherwise the rationale for the study is lost.
  • We assumed a comparison with onsite supported patients in order to better demonstrate the effectiveness of online support. We added a further figure (Figure 2) in order to show the comparison on the effectiveness of the psychological intervention performed onsite and online.

  • Personally, I would suggest relabelling the “IbIG” in “IG” and check for consistency throughout the Manuscript. The reader understands from the title that the intervention is internet-based and then the text becomes easier to read. Another aspect: If I rightly understand, the CG is a wait-list control group. This is very ethical and it is a strength of the paper. I would suggest stating this since the beginning in the abstract, i.e., “wait-list control group (CG)”, and being consistent throughout the Manuscript.
  • Having decided to keep the onsite supported group, to change the acronym IbIG to IG could create confusion. However, we agree with the reviewer on making the text more readable, and, for this reason, we have decided to change the acronym to I-IG. The correction was made throughout the text. According to the reviewer suggestion, in the Abstract and Materials and Methods sections, “wait-list control group (CG)” has been added.

  • A main aspect is the following: It is unclear if a mental health assessment was performed at baseline. Is there a T0 in the study, or the authors just assessed mental health outcomes at the end of the six months? In the case there was an assessment at T0 and then at T1, the T-tests chosen for the analyses might not be the best option, and Repeated Measures ANOVAs/MANOVAs might be a better solution. Please, clarify.
  • According to reviewer observation, we added in the subsection “Outcomes” the following sentence in order to clarify the evaluation process: “In all participants, a mental health assessment was performed at baseline and after six months of follow-up”. In the subsection “Statistical analysis”, “repeated measures analysis of variance (ANOVA)” has been added.

  • Figures and Tables are well-formatted and presented. Despite the paper is well written, I suggest having it revised by a native English speaker, as they could help make the paper easier to read.
  • According to reviewer suggestion, we obtained a review of the manuscript from a native English speaker.

  • Major revisions:
  • Abstract

Lines 17-19: “In a 6-month follow-up, participants in the IG demonstrated significant improvements in terms of PTSD risk (…), depression (…), and anxiety (…) if compared to the CG.

Lines 19-21: I would remove this sentence and these analyses, due my concern about the fact that these participants are not supported via internet, but rather in person.

  • The first correction has been made, according to reviewer suggestion. We have not removed the lines 19-21 for the reasons expressed in the point 2.

  • Keywords

The authors included some keywords that are already in the title, of course they are free to choose their keywords, but I would suggest excluding words already present in the title and include other concepts, e.g., stress, post traumatic stress disorder, ptsd, mental health.

  • We correct the keywords as the reviewer suggested.

  • Introduction

Line 54-55: Could the authors provide an English translation for “In Ascolto” in brackets?

  • According to reviewer suggestion, we added “Tune in” in brackets.

  • Materials and methods

Study sample: The authors did not describe the sample in terms of age range, mean age and standard deviation, gender, … Please, provide further details.

Lines 72-75: What this background add to our scientific knowledge? Please, outline this point.

  • About study sample, we just added the aforesaid information in Results section. Considering the lines 72-75, we think that cultural background is important, in the sense that all cultures are different even though they have that of answer the typically human question of giving meaning to experience, either individual and collective. We must also consider that every treatment process is not neutral, but culturally marked.

  • Statistical analyses

The description of statistical analyses is not in a logical order. I think the authors should start with:

Descriptive analyses of the study sample (e.g., age, gender frequency), then the authors do not need to report this step in “Statistical analyses”.

Analysis of distribution of the outcome variables, and report this at the beginning of “Statistical analyses”, as normally distributed data allow for further parametric tests to be undertaken.

Then the authors choice of T-tests is well described and suitable for this kind of analysis.

  • We ordered the Statistical analyses description as reviewer suggested.

  • Discussion

Lines 163-172: This paragraph should be included in the results section.

Generally, the discussion should be expanded.

  • The aforesaid paragraph is a summary of the results obtained in order to compare with other published studies. The Discussion section has been improved also with other references according to reviewer observation.

  • Minor revisions:
  • Abstract

Lines 10-11: “an increasing demand”

Line 14: missing space “7-item were”

Line 15: remove been “was administered”

Line 23: please, remove “even”, “can be improved by internet-based psychological intervention”

  • All minor revisions have been made.

  • Introduction

Line 29: please, remove “virus”, “a novel Severe Acute Respiratory Syndrome Coronavirus 2 (SARS-CoV-2) emerged worldwide.”

  • All minor revisions have been made.

  • Materials and methods

Lines 98-99: please, add spaces “anxiety disorder along with…”

  • All minor revisions have been made.

Reviewer 2 Report

Thank you for allowing me to review this paper. I reviewed it with great interest, as it is current and vital.

General comment: the article needs a more in-depth bibliographic review connecting the topic mainly with mental health during SARS-Cov-2 Pandemic. Intruduction chapter is insufficient. Coivd19 has impacted both social&economic policies, populations, policy makers and healthcare systems since early 2020. We still haven't found every single answer, but the bibliography is wide and easy to access. 

To be more precise:
1. Please consider official confirmation while writing about the beggining of the Covid19 (line 29, line 32 and line 34). For now it's quite clear and obvious, but for the future readers some facts could be useful.

2. (line 36) When it's written "In current studies..." please provide apropriate sources (i.a.  https://doi.org/10.1111/inm.12726 or https://doi.org/10.1192/j.eurpsy.2021.222 and more)

3. (line 39) When it's written "Psychological intervcention has a positive impact on (...) in health care workers and patients." it should be based on the litarature of the subject. Please provide adequate bibliography (i. a.. https://doi.org/10.1038/s41581-020-0314-5, https://doi.org/10.1093/geroni/igab046.1581 )

4. (line 49) Please consider a short descripiton about Bauerle et al. A few sentences would raise the value of the paragraph.

8. Please be more precise within the line 54. What does exactly mean "in the light of the current health emergency"? Again, it's important, mainly for the future readers.

9. (line 187) Please provide apropriate reference list when it's stated "to our knowledge, few studies have examined online psychological intervention ..."

I'd like to underline, enclosed manuscript need more in-depth bibliographic review.

[1] Please consider additional information about CG. What kind of help/solutions were implemented? 

[2] A part of the study related to technological tools should be extended. One short paragraph (between the lines 180-182) is not enough. In particular, when it's written "An innovative approach of this study is..."

A minor typo errors:

1. While building the sentence within lines 45-46 something went wrong.

2. Words in the line 123 need to be separated.

Author Response

  1. Thank you for allowing me to review this paper. I reviewed it with great interest, as it is current and vital. General comment: the article needs a more in-depth bibliographic review connecting the topic mainly with mental health during SARS-Cov-2 Pandemic.
  1. We thank the reviewer for his/her positive and constructive observations. Below, please find item-by-item responses to your comments, which are included verbatim.

  1. Intruduction chapter is insufficient. Coivd19 has impacted both social & economic policies, populations, policy makers and healthcare systems since early 2020. We still haven't found every single answer, but the bibliography is wide and easy to access.

2. The Introduction section has been improved according to reviewer suggestions.

  1. To be more precise:

3.1. Please consider official confirmation while writing about the beggining of the Covid19 (line 29, line 32 and line 34). For now it's quite clear and obvious, but for the future readers some facts could be useful.

3.1. According to reviewer observation, we deleted “On February 21, 2020” and re-write “On 2020”.

3.2. (line 36) When it's written "In current studies..." please provide apropriate sources (i.a.  https://doi.org/10.1111/inm.12726 or https://doi.org/10.1192/j.eurpsy.2021.222 and more).

3.2. We provided appropriate sources as recommended by the reviewer.

3.3. (line 39) When it's written "Psychological intervcention has a positive impact on (...) in health care workers and patients." it should be based on the litarature of the subject. Please provide adequate bibliography (i. a.. https://doi.org/10.1038/s41581-020-0314-5, https://doi.org/10.1093/geroni/igab046.1581 ).

3.3. We added the reference as reviewer suggested.

3.4. (line 49) Please consider a short descripiton about Bauerle et al. A few sentences would raise the value of the paragraph.

3.4. We explained the Bäuerle et al. study.

3.5. Please be more precise within the line 54. What does exactly mean "in the light of the current health emergency"? Again, it's important, mainly for the future readers.

3.5. According to reviewer observation, we changed in “In light of SARS-CoV-2 emergency”.

3.6. (line 187) Please provide appropriate reference list when it's stated "to our knowledge, few studies have examined online psychological intervention ...". I'd like to underline, enclosed manuscript need more in-depth bibliographic review.

3.6. We re-phrase the sentence in following way: “to our knowledge, studies that have examined online psychological intervention effectiveness for improving PTSD, depression and anxiety symptoms related to SARS-CoV-2 did not fully consider evaluating the usability of the technology”. Moreover, we improved Discussion section and added further references.

  1. Please consider additional information about CG. What kind of help/solutions were implemented?

4. According to the reviewer suggestion, in the Abstract and Materials and Methods sections, “wait-list control group (CG)” has been added in order to clarify.

  1. A part of the study related to technological tools should be extended. One short paragraph (between the lines 180-182) is not enough. In particular, when it's written "An innovative approach of this study is..."

5. According to the reviewer observation, we added other study references that support our approach.

  1. A minor typo errors:

6.1. While building the sentence within lines 45-46 something went wrong.

6.1. The sentence has been re-write as the reviewer observed.

6.2. Words in the line 123 need to be separated.

6.2. All typos have been correct.

Reviewer 3 Report

Some suggestions are presented, considering the analysis of the document:

In table 2, when referring to the educational level, it is not clear what it refers to, the value presented. Is it referring to the number of academic years?

One of the exclusion criteria was having a history of psychiatric disorders or taking psychiatric medications. However, in table 2, there are records of subjects (64) with psychological disorders. Will they not be related? Was it not considered an exclusion criteria?

In the item results, the results are presented descriptively, with the flowchart and tables referring to the characterization of the sample presented below. As a suggestion for improvement, I suggested that the results (flowchart and tables) be framed in the text (ex: description of results in table 2, followed by the respective table...)

Author Response

  1. Some suggestions are presented, considering the analysis of the document.

1. We thank the reviewer for his/her positive and constructive observations. Below, please find item-by-item responses to your comments, which are included verbatim.

  1. In table 2, when referring to the educational level, it is not clear what it refers to, the value presented. Is it referring to the number of academic years?

2. Yes, it is referring to academic years number. We added “in years” in table in order to clarify.

  1. One of the exclusion criteria was having a history of psychiatric disorders or taking psychiatric medications. However, in table 2, there are records of subjects (64) with psychological disorders. Will they not be related? Was it not considered an exclusion criteria?

3. According to reviewer observation, we re-phrase the word “psychiatric” with the word “psychotic” and deleted the sentence part “or taking psychiatric medications” in Materials and Methods section.

  1. In the item results, the results are presented descriptively, with the flowchart and tables referring to the characterization of the sample presented below. As a suggestion for improvement, I suggested that the results (flowchart and tables) be framed in the text (ex: description of results in table 2, followed by the respective table...)

4. We framed the results (flowchart and tables) in the text as suggested by the reviewer.

Round 2

Reviewer 1 Report

Second review of “Internet-based Psychological Interventions during SARS-CoV-2 pandemic: an Experience in South of Italy” version 2

The authors made a good work in improving the paper and enriched some parts of the introduction and the discussion, which now provide more meaningful insight on the implementation of internet-based psychological interventions with particular regards to the COVID-19 pandemic situation.

However, I still have two minor concerns that the authors should address before presenting their work to the scientific community and the general public. These regards to: (1) the analyses reported; and, more important, (2) the extendibility to other situations.

With regards to the analyses, when I look at the results from T-tests, I am fully convinced that the psychological intervention was positive and effective. However, with this paper the authors are presenting the results to the scientific community, and I feel there is still confusion in the way analyses are reported. In this second version, the authors mentioned repeated measures ANOVAs in the analyses description, but it does not seem that these analyses have been actually performed, or, at least, they have not been reported. I still think that repeated measures ANOVAs are the most appropriate test to evaluate differences between baseline and follow-up and between the three groups (I-IG, O-IG, and CG) and I encourage the authors to run and report these analyses.

Alternatively, the authors should remove “repeated measures ANOVA” from the analyses description (Section 2.2 “Outcomes”) and they could just present T-tests results. In this case, it may be logical to present in Table 1 descriptive statistics of the three groups (I-IG, O-IG, and CG), and then I would suggest merging Table 2 and Table 3, and report means of the psychological variables for the three groups in the same Table.

Along with this, I personally find data reported in Figure 2 not intuitive, and I would encourage authors to better explain this kind of analyses in the analyses description section and add some references to reinforce their choice. Moreover, this part of analyses is not discussed in the “Discussion” section. In my opinion, Figure 2 and the related analysis do not add anything to the paper, and this would still be suitable for publication without them. If the authors want to maintain them, I encourage them to better explain their meaning.

With regards to extendibility to other situations, I think the authors’ findings have many more practical implications than those outlined in the Discussion. For example, could internet-based interventions be useful also when the COVID-19 pandemic will be over? If so, in which cases? I think, there are many situations where patients are constrained at home or have difficulties in reaching a healthcare structure, e.g., disability, illness, injury, and in these case internet-based interventions may be very useful. Additionally, are there any personality characteristics that can make internet-based interventions more appropriate? I am thinking of introverted people, could these patients be more inclined to seek for internet-supported intervention rather than to go in a public structure? I know this is speculative, but I would encourage authors to express some of their thoughts and outline directions for future research in the Discussion.

I hope these comments will help the authors to further improve the Manuscript.

Author Response

  1. The authors made a good work in improving the paper and enriched some parts of the introduction and the discussion, which now provide more meaningful insight on the implementation of internet-based psychological interventions with particular regards to the COVID-19 pandemic situation. However, I still have two minor concerns that the authors should address before presenting their work to the scientific community and the general public. These regards to: (1) the analyses reported; and, more important, (2) the extendibility to other situations.
  1. We thank the reviewer for his/her positive and constructive observations. Below, please find item-by-item responses to your comments, which are included verbatim.

  1. With regards to the analyses, when I look at the results from T-tests, I am fully convinced that the psychological intervention was positive and effective. However, with this paper the authors are presenting the results to the scientific community, and I feel there is still confusion in the way analyses are reported. In this second version, the authors mentioned repeated measures ANOVAs in the analyses description, but it does not seem that these analyses have been actually performed, or, at least, they have not been reported. I still think that repeated measures ANOVAs are the most appropriate test to evaluate differences between baseline and follow-up and between the three groups (I-IG, O-IG, and CG) and I encourage the authors to run and report these analyses. Alternatively, the authors should remove “repeated measures ANOVA” from the analyses description (Section 2.2 “Outcomes”) and they could just present T-tests results. In this case, it may be logical to present in Table 1 descriptive statistics of the three groups (I-IG, O-IG, and CG), and then I would suggest merging Table 2 and Table 3, and report means of the psychological variables for the three groups in the same Table.
  1. According to the reviewer suggestions and observations, we deleted ANOVA from analyses description. Moreover, we merged Table 2 and Table 3 as suggested.

  1. Along with this, I personally find data reported in Figure 2 not intuitive, and I would encourage authors to better explain this kind of analyses in the analyses description section and add some references to reinforce their choice. Moreover, this part of analyses is not discussed in the “Discussion” section. In my opinion, Figure 2 and the related analysis do not add anything to the paper, and this would still be suitable for publication without them. If the authors want to maintain them, I encourage them to better explain their meaning.                                                                        3. We deleted Figure 2 and related analysis.

  1. With regards to extendibility to other situations, I think the authors’ findings have many more practical implications than those outlined in the Discussion. For example, could internet-based interventions be useful also when the COVID-19 pandemic will be over? If so, in which cases? I think, there are many situations where patients are constrained at home or have difficulties in reaching a healthcare structure, e.g., disability, illness, injury, and in these case internet-based interventions may be very useful. Additionally, are there any personality characteristics that can make internet-based interventions more appropriate? I am thinking of introverted people, could these patients be more inclined to seek for internet-supported intervention rather than to go in a public structure? I know this is speculative, but I would encourage authors to express some of their thoughts and outline directions for future research in the Discussion.  4. In Discussion section, we added the following sentences in order to clarify our thought: “Presumably, when the pandemic is over, the normality to which you will return will coincide anyway with a new way of work. The new modes will not replace tout court the classic setting, but they will integrate to it. It will therefore be necessary to assess case by case if the request to remain online can be evolutionary or defensive. In the prospect of greater integration of the instrument should be assessed on a case-by-case basis who can benefit clinically and who not, also with regard to any management entirely at a distance. This is a subject of interest, growing which is worth continuing to reflect”.

  1. I hope these comments will help the authors to further improve the Manuscript.                                                                                                    5. We thank the reviewer again for his suggestions that have certainly improved and enriched our work.